# Synthetic Strategies of Pyrimidine-Based Scaffolds as Aurora Kinase and Polo-like Kinase Inhibitors

**DOI:** 10.3390/molecules26175170

**Published:** 2021-08-26

**Authors:** Mrunal Jadhav, Kaksha Sankhe, Richie R. Bhandare, Zehra Edis, Samir Haj Bloukh, Tabassum Asif Khan

**Affiliations:** 1Department of Pharmaceutical Chemistry and Quality Assurance, SVKM’s Dr. Bhanuben Nanavati College of Pharmacy, Mumbai 400056, India; mrunal.jadhav@yahoo.in (M.J.); kakshavs@gmail.com (K.S.); 2Department of Pharmaceutical Chemistry, College of Pharmacy & Health Sciences, Ajman University, Ajman P.O. Box 346, United Arab Emirates; r.bhandareh@ajman.ac.ae; 3Center of Medical and Bio-Allied Health Sciences Research, Ajman University, Ajman P.O. Box 346, United Arab Emirates; s.bloukh@ajman.ac.ae; 4Department of Clinical Sciences, College of Pharmacy & Health Sciences, Ajman University, Ajman P.O. Box 346, United Arab Emirates

**Keywords:** synthesis, aminopyrimidines, aurora kinase, polo-like kinase, anticancer

## Abstract

The past few decades have witnessed significant progress in anticancer drug discovery. Small molecules containing heterocyclic moieties have attracted considerable interest for designing new antitumor agents. Of these, the pyrimidine ring system is found in multitude of drug structures, and being the building unit of DNA and RNA makes it an attractive scaffold for the design and development of anticancer drugs. Currently, 22 pyrimidine-containing entities are approved for clinical use as anticancer drugs by the FDA. An exhaustive literature search indicates several publications and more than 59 patents from the year 2009 onwards on pyrimidine derivatives exhibiting potent antiproliferative activity. These pyrimidine derivatives exert their activity via diverse mechanisms, one of them being inhibition of protein kinases. Aurora kinase (AURK) and polo-like kinase (PLK) are protein kinases involved in the regulation of the cell cycle. Within the numerous pyrimidine-based small molecules developed as anticancer agents, this review focuses on the pyrimidine fused heterocyclic compounds modulating the AURK and PLK proteins in different phases of clinical trials as anticancer agents. This article aims to provide a comprehensive overview of synthetic strategies for the preparation of pyrimidine derivatives and their associated biological activity on AURK/PLK. It will also present an overview of the synthesis of the heterocyclic-2-aminopyrimidine, 4-aminopyrimidine and 2,4-diaminopyrimidine scaffolds, and one of the pharmacophores in AURK/PLK inhibitors is described systematically.

## 1. Introduction

Cancer has been the leading cause of death in recent years. The occurrence of many cancers is due to discrepancies in the cell cycle or cell division during the mitotic phase. This imbalance is triggered due to misalliance of chromosomes resulting in aneuploidy and incomplete genes [1,2,3]. A certain group of kinases such as polo-like kinases (PLK) and aurora kinases (AURK) play an important role in chromosome segregation during mitosis. AURK and PLK are a group of serine (Ser)/threonine (Thn) kinases that control mitosis [4,5]. Aurora kinase has three subtypes: aurora kinase A (AURKA), B (AURKB), and C (AURKC) [6,7,8], whereas PLK has five subtypes (PLK-1 to PLK-5) in human cells [9,10,11]. AURK and PLK play a central role in regulating mitotic entry and progression as they are associated with centrosome maturation, mitotic spindle formation and chromosome separation [12,13]. PLK is overexpressed in ~80% of human tumor types and is not often expressed in normal tissues [14,15]. AURK and PLK along with cyclin-dependent kinase (CDK) and various adaptor proteins cooperate in various spatiotemporal context to regulate mitosis and cytokinesis. (Figure 1) [16]. 

Both AURKA and PLK along with various cofactors regulate cell entry and smooth progression through mitosis. Given its central role in mitosis, the AURKA/PLK1 axis is associated with many types of cancers. In addition, both AURK and PLK prevent centrosome maturation and double spindle formation [17]. AURK is required for preliminary inactivation of PLK that delays mitotic arrest giving rise to unipolar spindle bodies thereby inducing apoptosis [18,19]. Recent literature indicates that the AURKA/PLK1 axis promotes cancer cell growth and survival independently from its well-established role in mitosis by phosphorylating noncanonical substrates, many of which are of significant interest in hematologic malignancies, including “high-risk” lymphoproliferative disorders [20,21]. Thus, a novel drug design strategy utilizing a combination of multiple targets has attracted great attention recently. The first clinical trials on AURK inhibitor were reported in 2005 along with 70 clinical trials in various phases published till date to evaluate the clinical efficacy of the molecules developed as AURK inhibitors [22]. These protein kinases (AURK and PLK) catalyse the transfer of the γ-phosphate group of ATPs to substrate containing Ser/Thr amino acid residues. Many studies have suggested that kinase is one of the most promising targets in cancer therapeutics [23]. AURKA and PLK-1 catalytic domains have been identified via screening of natural and synthetic compound libraries. These inhibitors interfere with catalytic activity and diminish its expression. Many of these inhibitors are in different phases of clinical trials (Figure 2) [16,24]. Various molecules have been investigated for their utility as AURK and PLK inhibitors. Around 50% of new entities in clinical trials possess pyrimidine-fused scaffolds such as 2-aminopyrimidines, 2,4-diaminopyrimidines, and 4-aminopyrimidines. Markedly, compounds bearing these scaffolds exhibited potent inhibitory activity. There are about 100 reports on novel AURK inhibitors in the last 20 years. Many small molecules have been developed and synthesized as AURK inhibitors with encouraging results in cytotoxicity studies. The most potent molecules that translated to clinical trials such as alisertib (MLN8237), barasertib (AZD1152), ENMD-2076, AMG900 and PF-03814735 are pyrimidine derivatives. They exhibited potent inhibition of AURK with IC_50_ of 0.0012 µM (AURKA), 0.00037 µM (AURKB), 0.014 µM (AURKA), 0.004 µM (AURKB) and 0.0008 µM (AURKA) respectively [25,26,27,28,29,30]. Pyrimidine derivatives such as BI2536, BI6727, DAP-81 showed potent inhibition of PLK with IC50 of 0.00083 µM, 0.00087 µM and 0.0009 µM respectively [31]. These AURK and PLK inhibitors act as ATP competitive inhibitors (Figure 2). A comprehensive understanding of the molecular constraints of the ATP-binding site of AURK and PLK is an essential step in designing new inhibitors of this subfamily of kinases. Most of the AURK and PLK inhibitors possess adenine-like scaffolds and showcase similar binding modes like formation of hydrogen bonds between kinase and the inhibitor. Pyrimidine-fused compounds such as 2-aminopyrimidines, 2,4-diaminopyrimidines, and 4-aminopyrimidines can form hydrogen bonds with amino acid residues in the hinge region of these kinases [32]. This article is an effort to present an overview of the synthetic strategies of pyrimidine derivatives along with their biological activity on AURK/PLK. This review envisages a discussion on synthesis of heterocyclics-2-aminopyrimidines, 4-aminopyrimidines and 2,4-diaminopyrimidines and present a systematic description on one of the pharmacophores of combinatorial AURK/PLK inhibitors.

## 2. Synthetic Strategies of Pyrimidines

### 2.1. 2-Aminopyrimidines

2-Aminopyrimidine-based derivatives exhibit a broad spectrum of activities, hence, the synthesis of these compounds has attracted attention for many decades, and the development of new methods for the synthesis of these compounds continues to be of great interest. The synthesis of substituted pyrimidines involves two general methodologies. One method involves condensation of moieties with required substituents to get the heterocycle. Another method involves replacement of the substituent at position 2 of the pyrimidine ring with an amino group. The second method is less efficient and gives target products in low yields, particularly, in the reactions with arylamines, that require a large excess of nucleophiles [33].

### 2.2. Condensation Reactions

The preparation of 2-aminopyrimidines is most often based on convergent synthesis, which is often referred to as conventional synthesis. This method (Figure 3) is based on the condensation of dielectrophiles containing three-carbon chain and carbonyl functionality, ester, or nitrile group with dinucleophiles containing the N-C-N moiety. If guanidine is used as the dinucleophilic component, the formation of pyrimidine ring is accompanied by the introduction of amino group at position 2 as described in Section 2.2.1. 2-Aminopyrimidines substituted at the amino group are formed in the reactions with alkyl-, aryl-, or arylsulfonylguanidines, and with dicyanodiamide, mentioned in Section 2.2.2 [33].

#### 2.2.1. Synthesis of 2-Aminopyrimidines from Guanidine and β-Dicarbonyl Compounds

β-Dicarbonyl compounds and their derivatives (β-ketoesters, β-ketonitriles, and acid chlorides, whose terminal functional fragments can react with nucleophiles), serve as the dielectrophilic component. The condensation is performed in polar solvents with heating in presence of a condensing agent; in some cases, the reactions are performed by fusion. The yield of 2-aminopyrimidines containing various substituents in the ring (Alk, Ar, Het, CF_3_, CHF_2_, OH, CO_2_Et, CN, NHCOAr) ranges from 60–95% [33]. 

Bayramoğlu et al. accomplished the synthesis of 2-aminopyrimidines **3a–f** (Scheme 1) by performing the corresponding reactions under both conventional and ultrasonic irradiation conditions. Conventional methods of pyrimidine synthesis have long reaction times and the crude product yields vary from 54–78% [34,35,36,37], while the ultrasound-assisted cyclization reactions facilitate the formation of target molecules in less than 1 h. Strong bases such as NaOC_2_H_5_ were used in several experiments giving final yields close to those of the conventional method. Numerous experimental parameters were explored in an effort to optimize the synthetic procedures. These involved selection of suitable base (Na_2_CO_3_, NaOH, and NaOC_2_H_5_) and molar ratio of the starting material to achieve the desired cyclization reaction. It was observed that Na_2_CO_3_ is adequately reactive to act directly for the cyclization reaction of 2-amino-4,6-dimethylpyrimidine and 2-amino-4-hydroxy-6-methylpyrimidine. A stronger base (NaOC_2_H_5_) is required for the synthesis of 2-amino-4,6-dihydroxypyrimidine and its derivatives containing different alkyl or alkylidene groups at the 5-position (Table 1). Additionally, the molar ratio of the respective substrate to base were determined, different ratios of base proportion versus 2-aminopyrimidine derivatives were evaluated and significantly better results were attained by using 1:1 equivalents. In the conventional synthesis of 2-aminopyrimidines, it is necessary to keep the temperature around 100 °C for about 5–8 h, whereas in the ultrasonic-assisted reactions temperature range of 60–70 °C is sufficient and the reaction time period is reduced to 30 min [38].

**Where**,

**Table 1 molecules-26-05170-t001:** Derivatives of 2-aminopyrimidine.

Sr. No.	R_1_	R_2_	R_3_	R_4_	R_5_
**3a**	CH_3_	CH_3_	H	CH_3_	CH_3_
**3b**	CH_3_	OC_2_H_5_	H	CH_3_	OH
**3c**	OC_2_H_5_	OC_2_H_5_	H	OH	OH
**3d**	OC_2_H_5_	OC_2_H_5_	C_2_H_5_	OH	OH
**3e**	OC_2_H_5_	OC_2_H_5_	C_4_H_9_	OH	OH
**3f**	OC_2_H_5_	OC_2_H_5_	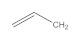	OH	OH

#### 2.2.2. Synthesis of 2-Aminopyrimidines from α, β-Unsaturated Ketones

Dimethylamino- and alkoxy derivatives of α,β-unsaturated ketones **4**, viz., vinylogues of amides (X = NMe_2_) and enol ethers (X = OAlk), also serve as efficient dielectrophiles in the synthesis of 2-aminopyrimidines. The condensation of derivatives **5** with alkyl(aryl)guanidines under reflux in ethanol affords the corresponding alkyl-, aryl-, or hetaryl-substituted 2-aminopyrimidines (Scheme 2). In this condensation, cyclic ketones give carbo- (C5–C12) or heterocycle fused 2-aminopyrimidines in 35–75% yields [39].

### 2.3. Substitution Reaction

2-Aminopyrimidines were synthesized by replacing halogen atom in mono- (**6**) and dichloropyrimidine (**9**) mostly with the use of ammonia, primary aliphatic and aromatic amines, secondary acyclic and cyclic amines.

This method, which was modified by the use of modern reagents (bases and catalysts), is widely used in the synthesis of biologically active arylaminopyrimidines providing a diversity of structures for screening. Amination of 2-chloropyrimidine is usually carried out in presence of palladium catalyst under high pressure [40,41]. However, dichlorides **9a,b** are aminated by aromatic amines mostly at position 4 rather than at position 2 (Scheme 3) [42,43,44,45].

The amination of 2,4-dimethoxy-5-nitropyrimidine (Scheme 4) (11) affords the corresponding 2-aminopyrimidine in moderate yield [46].

The replacement of a hydrogen atom at position 2 of the pyrimidine ring by the amino group occurs by the addition of the nucleophile, ring opening, and ring closure (ANRORC) mechanism, whereas the amination at position 4 occurs without the closure of the pyrimidine ring. The replacement of the chlorine atom in 2-chloro-4-phenylpyrimidine in the KNH_2-_NH_3_ system at −33 °C also occurs by the ANRORC mechanism [47]. 2-Alkylaminopyrimidines can be synthesized by aminolysis (with ammonia, butyl-, pentyl- and cyclohexylamines) of 2-alkoxy-, 2-alkylthio- **13**, methyl(phenyl)sulfinyl-, or -sulfonylpyrimidines **14** (Scheme 5).

Sulfoxides and sulfones proved to be slightly more reactive and alkoxy and alkylthio derivatives are much more reactive in amination reactions compared to the corresponding chloropyrimidines. However, it is reasonable to use these compounds as starting reagents when the corresponding halogen derivatives are unavailable or unstable on storage [48]. 4,6-Diarylpyrimidine-2-thiones **16** react with guanidine to form 2-aminopyrimidines (Scheme 6).

The method of synthesis of aminopyrimidines substituted at the exocyclic nitrogen atom from available 2-(or 4-, 6-)-aminopyrimidines by heating salts of the starting substrates with amines did not gain acceptance because the transamination virtually does not occur at position 2 of the ring. 2-R-Substituted aminopyrimidines can be synthesized by direct alkylation of 2-aminopyrimidine on prolonged heating [33].

### 2.4. 4-Aminopyrimidines

4-Aminopyrimidines are main intermediates in the synthesis of various molecules such as the well-known vitamin B_1_ (thiamine), the natural thiamine antagonist bacimethrin, or the chemotherapeutic agent trimethoprim. Several synthetic procedures are available for the synthesis of 4-aminopyrimidines.

### 2.5. Synthesis of 4-Aminopyrimidines

Williams et al. discovered a four-step synthesis scheme for making 4-aminopyrimidine units using ethyl 3-ethoxypropionate as the starting material (Scheme 7). However, corrosive and toxic reagents like ammonium gas and phosphoryl chloride were utilized to get satisfactory yields [49]. Karad et al. developed a metal-catalyzed or strong acid-mediated intermolecular cycloaddition of ynamides with nitriles for the synthesis of 4-aminopyrimidines using Scheme 8 [50]. Baxendale et al. described the microwave synthesis of 4-aminopyrimidines using single nitriles in the presence of a catalyst such as potassium *tert*-butoxide (Scheme 9). The reaction of acetonitrile in 3 mol of potassium tert-butoxide gave 2,6-dimethylpyrimidin-4-ylamine with a yield of 48% after 20 min at 140 °C. The synthesis of the final product was confirmed using liquid chromatography-mass spectroscopy (LC-MS) and the results indicated 90% conversion [51]. Zhu et al. described a catalyst-free and facile synthetic scheme for the creation of 4-aminopyrimidine scaffold. The β-enaminonitriles can be converted to 4-aminopyrimidines through modifiable reaction conditions (Scheme 10). β-Enaminonitriles play a critical role in the preparation of heterocyclic compounds that are extensively synthesized in dyes, pesticides, medicines, and fungicides. Hence, economical and practical synthetic routes of 4-aminopyrimidines and β-enaminonitrile has immense utility in organic chemistry and pharmaceutical chemistry. The application of organonitriles as a supplemental source of nitrogen is an area of great scientific interest currently in the synthesis of 4-aminopyrimidine moieties [52]. 

Zhu et al. improved the reaction conditions by modifying the time, the ratio of the base and substrate. The reaction of benzonitrile (0.6 mmol) and 3-phenylpropionitrile (0.2 mmol) was carried out at 120 °C for 24 hrs in the presence of the base lithium hexamethyldisilazide (0.2 mmol) (LiHMDS) in dimethoxyethane (DME) solvent resulting in a 90% yield of 5-benzyl-2,6-diphenyl-4-aminopyrimidines. This study showed that LiHMDS salt base is effective for deprotonating α-H on a 3-phenylpropionitrile substrate. The imine intermediate was formed via attacking the C-atom of the cyanide group of benzonitrile by deprotonated anion methylene carbon. Besides, organic bases and alkali salts like NaOH, CS_2_CO_3_, EtONa, tBuOK, 1,8-diazabicyclo [5.4.0]-undec-7-ene (DBU), and 4-dimethylamino pyridine (DMAP) formed a final product, i.e., 5-benzyl-2,6-diphenyl-4-aminopyrimidine in much lesser yields. Polar solvents were found to be unfavorable for conducting the reaction because hydrogen bonding interactions between the intermediate amines and polar solvents might be constraining the synthesis of 4-aminopyrimidine scaffold. A better yield of 5-benzyl-2,6-diphenyl-4-aminopyrimidine may be achieved with the use of ethers like tetrahydrofuran (THF), *p*-methoxyphenyl ether (MeOPh), and DME. Of these, DME gave a substantial yield at 100–120 °C, while a lower temperature of 40 °C might reduce the product yield. An additional trial was directed by Zhu and colleagues using a mixture of benzonitrile and β-enaminonitrile. This mixture was allowed to mix for 24 h under suitable conditions for synthesizing 4-aminopyrimidines with a yield of 90%. Various other routes have been explored for the synthesis of substituted 4-aminopyrimidines using 2-thiophenecarbonitrile, benzonitrile, and 4-bromobenzonitrile as electron acceptors and 3-phenylpropionitrile as an electron donor. The resulting yields of the final products comprising 4-aminopyrimidines were in the range of 70% to 60%, respectively [52]. Jachak et al. reported the synthesis of 4-aminopyrimidines in 80% yield by heating substituted benzoylacetonitriles and substituted ureas at 70 °C (Scheme 11) [53]. Letinois et al. described a synthetic scheme (Scheme 12) for the synthesis of 4-aminopyrimidines through by condensation of β-cyanoenolates with amidine hydrochlorides. This route is valid for the production of 4-aminopyrimidines at an industrial scale. β-Cyanoenolates and amidine hydrochloride (starting materials) are freely soluble in dimethyl sulfoxide (DMSO), ethanol, methanol, and dimethylformamide (DMF). However, both these starting materials decompose in the presence of these solvents. With other solvents such as ethers and esters, the solubility of the starting material was less. β-Cyanoenolates and amidine hydrochloride in presence of solvents like dioxane, 3-pentanone, or toluene form sticky greyish brown color suspensions on heating at 75–100 °C and also get trapped to the sidewall of the flask, giving a 37% yield. They confirmed the synthesis of pyrimidines by using high-performance liquid chromatography (HPLC). DSC analysis indicated that decomposition of the enolates starts at 115 °C. Other additives (catalysts) such as H_2_SO_4_ were added to the reaction mixture to increase the yield of the product and this increased the yield to 45%. However, the advantage of this additive was somewhat insufficient compared to the uncatalyzed experiment (37% yield). Ammonium chloride (NH_4_Cl, a Brönsted acid) had undesirable effects on the formation of the product (13% yield), and sodium methoxide NaOCH_3_ (a Brönsted base) did not form a final product. The incorporation of alanine into the reaction mixture yielded 40% of product while the addition of triethylamine (Et_3_N) in toluene yielded 42%. The addition of Lewis acid-like aluminum trichloride (AlCl_3_) and gallium trichloride (GaCl_3_), decomposed the starting material whereas Lewis acids such as titanium dioxide (TiO_2_) and titanium tetrachloride (TiCl_4_) formed a final product with a better yield of 55% and 52%. Aminophilic Lewis acids were found to be more effective as catalysts. Iron chloride (FeCl_2_) enhanced the yield of the final product to 60% [54].

The incorporation of salts like dichloro(1,5-cyclooctadiene)ruthenium (II) (Ru-(COD) Cl_2_), and cobalt chloride (CoCl_2_) formed 4-aminopyrimidine in 72% and 70% yield, respectively, whereas with the addition of copper chloride (CuCl_2_) the yield was 87%. The addition of pentahydrate or dihydrate of copper chloride (CuCl_2_) and copper sulfate (CuSO_4_) yielded 63% and 60%, respectively. Zinc halides such as zinc bromide (ZnBr_2_), and zinc chloride (ZnCl_2_) were also evaluated as catalysts, resulting in 74% and 80% yield, respectively. In this trial, ZnCl_2_ and CuCl_2_ were the preferred Lewis acids for the synthesis of 4-aminopyrimidines with 80% and 87% yield, respectively. Several reaction solvents that could avoid the sticking issue and decomposition of the starting material were evaluated. Using solvents such as DMSO, DMF and NMP the sticking issue is resolved, but the reactants undergo decomposition. The authors did a comprehensive evaluation of the solvents regarding their physicochemical properties. A mixture of isopropanol and toluene was an appropriate solvent for experimenting as it limited the decomposition of the reactants. Also, the amidine hydrochloride and ZnCl_2_ stoichiometry affects the stirrability of the mixture. A mixture of 1.4 equiv of amidine hydrochloride with 0.15 equiv of ZnCl_2_ had non-stirrability issues, while 1.1 equiv of amidine hydrochloride with 0.2 equiv of ZnCl_2_ solved the stirring problem in isopropanol and toluene solvent [54].

### 2.6. Synthesis of 2,4-Diaminopyrimidines

Long et al. developed a synthetic scheme for synthesizing 2,4-diaminopyrimidines using Scheme 13. The coupling of 3-amino-5-methyl pyrazole with 2,4,6-trichloropyrimidine formed a C-4 substituted pyrimidine. The intermediate was formed using nucleophilic substitution of various anilines at the C-2 position of the pyrimidine core, and yields of the products ranged from 10–50% [55].

Ma et al. designed and synthesized 2,4-diaminopyridime scaffolds using 2,4-dichloride-5-fluoropyrimidine or 2,4-dichloride-5-nitropyrimidine as shown in Scheme 14. First the C4-chlorine of the pyrimidine ring in 2,4-dichloro-5-fluoropyrimidine or 2,4-dichloro-5-nitropyrimidine was swapped with several amines under various reaction conditions to produce the intermediate 2-chloro-4-amino-5-substituted pyrimidine derivatives. These intermediates were then refluxed with 4 N hydrochloric acid (HCl) in isopropanol or butanol to yield the desired compounds [56].

### 2.7. AURK and PLK Inhibition Studies of Aminopyrimidines

Long et al. developed a series of 12 N-trisubstituted pyrimidine scaffold derivatives (Scheme 13, Table 2) per Scheme 14 as potential AURK inhibitors. The most potent compound **38j** displayed AURKA and AURKB inhibition with IC_50_ values of 0.0071 and 0.0257 µM, respectively. The anti-proliferative activity of **38j** was tested on the U937 leukemia cell line with an in-vivo study of a xenograft nude mice model. The IC_50_ was 0.012 µM and it was indicated that **38j** repressed the tumor growth by 50–60%. In this study, a **38j** treated group of mice displayed no reduction in their body weight, indicating the low toxicity of **38j** [55]. 

Ma et al. synthesized analogues of nitroxide-labelled pyrimidines (Table 3) using the reactions shown in Scheme 14. Of these, **41l** was effective on many cancer cell lines in the anti-proliferative study with IC_50_ values of 0.89, 2.27, 11.41, and 5.73 on A-549, HeLa, LoVo, and HepG2, respectively. Almost all the analogues were perceived to be more potent than the VX-680 standard with **41l** being the most effective. **41l** exhibited AURKA and AURKB inhibition in a kinase-Glo-luminescent assay with IC_50_ values of 0.0093 and 0.0028 µM, respectively. **41l** was tested for immunofluorescence effects at two concentrations i.e., 2.5 and 5.0 µM in HeLa cell line on AURKA with Thr288 and AURKB with Thr232. These studies established that **41l** repressed AURKA autophosphorylation in a dose-dependent manner with slightly more selectivity towards AURKB. **41l** displayed AURKB inhibition at 2.5 µM and AURKA inhibition at 5.0 µM [56]. 

Takashi et al. synthesized compound **45** and its analogues as potential PLK inhibitor and patented this work (WO-2008081910-A1). Scheme 15 depicts the synthesis of compound **45** consisting a 2-aminopyrimidine moiety wherein R_1_ and R_2_, which may be the same or different, are each a hydrogen atom, a halogen atom, a lower alkyl group which may be substituted, or a cyclopropyl group; one of R_3_ and R_4_ is a hydrogen atom, while the other one of R_3_ and R_4_ is a lower alkyl group substituted with NR_3_R_b_, where R_3_ and R_b_ may be the same or different, are each a hydrogen atom, a lower alkyl group, a benzyl group, or a cycloalkyl group having three to six carbon atoms with one or more same or different substituents and R_5_ is a hydrogen atom, a cyano group, a halogen atom, or a lower alkyl group. Measurement of the inhibitory effect of the synthesized compounds was performed against PLK 1 activity and PLK 1 at cellular level using human uterine cervix cancer cell lines HeLaS3 cells. Of the 38 analogues synthesized compound **45a** showed better PLK-1 inhibitory activity with an IC_50_ of 1.8 nM and EC_50_ of 2.9 nM in a HeLaS3 cell line proliferation assay [57]. 

## 3. Conclusions and Future Prospects

Numerous studies have indicated the critical role of AURK and PLK in cancer cell proliferation. Recent reports suggest overexpression of AURK and PLK in a variety of cancers signifying their role as oncogenes in tumorigenesis and resulting in growing interest in these kinases for drug development in cancer therapeutics. There are several reports on small molecule AURK and PLK inhibitors that influence and constrain the downstream pathways of these kinases, subsequently controlling cancer progression. A variety of hybrid molecules comprising diverse scaffolds have been designed and synthesized with some translating into clinical trials. The recent years have witnessed intensive investigation in the development of AURK and PLK inhibitors accompanied by FDA drug approvals of the most promising entities. The existing chemotherapeutic management of cancer is accompanied by severe side effects and emerging resistance to current clinically used anticancer drugs. These limitations have provided impetus and has emerged as the chief driver of research programs involved in developing better and more effective drugs that can evade resistance with several molecules demonstrating good AURK and PLK inhibition. About 50% of new entities of AURK and PLK inhibitors in clinical trials possess 2-aminopyrimidine, 2,4-diaminopyrimidine and 4-aminopyrimidine scaffolds. Molecules bearing these scaffolds that have demonstrated effective AURK inhibition include alisertib (MLN8237), barasertib (AZD1152), ENMD-2076, AMG900 and PF-03814735 with IC_50_ of 0.0012 µM (AURKA), 0.00037 µM (AURKB), 0.014 µM (AURKA), 0.004 µM (AURKB) and 0.0008 µM (AURKA), respectively. Pyrimidine derivatives such as BI2536, BI6727, DAP-81 have showed potent inhibition of PLK with IC_50_ values of 0.00083 µM, 0.00087 µM, and 0.0009 µM, respectively. Molecules that exhibit significant inhibition of AURK and PLK specifically contain an adenosine mimicking scaffold such as a pyrimidine. Of these, N-trisubstituted pyrimidines **38j** and **45a** and nitroxide-labelled pyrimidines like **41l** showed better AURKA, PLK1 and AURKB inhibition with IC_50_ of 0.0071 µM, 1.8 nM and 0.0028 µM, respectively. A recent paper by Chi et al. reports the design and synthesis of a series of pyrimidine-based derivatives as AURKA inhibitors using structure-based drug design. These were designed to inhibit the proliferation of high-MYC expressing (MYC family oncogenes) expressing small-cell lung cancer cell line. One of the compounds in this series reduced cMYC and MYCN levels by >50% at 1.0 μM. [58] These scaffolds thus exhibited sufficient evidence making them attractive molecules for further cancer drug development research. Investigators have published numerous synthetic strategies for synthesizing pyrimidine derivatives. Analysis of the structures of therapeutically active synthetic 2-aminopyrimidine derivatives indicated that the presence of aniline moiety, amide bond, and halogen atoms, along with heterocyclic substituents (piperazine, pyridine, imidazole), plays an important role in activity. 2-Aminopyrimidine is the most significant moiety accountable for the biological activity of its analogs. This is true not only for substituted 2-aminopyrimidines, but also for derivatives with fused rings, including pteridines, pyridopyrimidines, imidazopyrimidines and purines. A safe, efficient, and facile scheme to produce substituted 4-aminopyrimidines by using adjustable organonitriles as starting materials is reported. The various synthetic strategies that have been discussed in the article are more with mixed nitrile, simple work-up, scale-up, and broad substrate scope. The involvement of easily available starting materials, outstanding functional group acceptability, and mild reaction conditions make these synthetic procedures applicable for the synthesis of pyrimidines. This effort might provide evidence for synthesizing useful multi-substituted aminopyrimidine molecules that can be translated into in vivo studies and subsequently into clinical trials.

## Data Availability

Not applicable.

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
