# Peer review of "Synthetic Strategies of Pyrimidine-Based Scaffolds as Aurora Kinase and Polo-like Kinase Inhibitors"

_molecules, 2021, doi:10.3390/molecules26175170_

Round 1
Reviewer 1 Report
Please see the attached file.

Author Response
Please see the attachment.
We have revised the manuscript as per the Reviewer 1 comments and have attached the detailed response herewith.
We have done this to the best of our capability, however please let us know if we have to further strengthen this manuscript.
We appreciate your critique as it has helped us to modify the manuscript to make it technically stronger.

Reviewer 2 Report
To be clear and simple, I do not recommend the publication of the review paper 'Design and Synthetic Strategies of Pyrimidines as Aurora Kinase and Polo like Kinase Inhibitors '.
I have the following points,
1, In this paper, the author mainly described the synthetic strategies of pyrimidines. There is no related information of the design of the inhibitor of Aurora Kinase and Polo like Kinase. Thus, this title is not well correlated with the content, and this causes the misleading information to the audience.
2, Being of an important fragment, pyrimidines was frequently existed in the structures of various inhibitor of Aurora Kinase and Polo like Kinase. However, this is only the simple fact, the author did not analyze to explore the binding mode between this fragment and the enzyme. However, the authors over exaggerated the fragment of pyrimidine, and called it inhibitors of the enzyme. This is not correct.
3, The format of the abstract and the conclusion part are very messy. They need to be rewritten.
This is only a preliminary draft, it need to be well revise to make it a paper.
Thus, I do not recommend the publication of this paper.
Author Response
Please see the file attached.

Round 2
Reviewer 1 Report
The revision is sufficient for the publication.
Author Response
The reviewer commented that the MS has been revised sufficiently for publication.
Response- We appreciate this
Point 1- English language and style are fine/Minor spell check required.
Response 1- The MS has been revised and spelling errors have been addressed to the best of our ability.
Reviewer 2 Report
After read through the whole manuscript and the cover letter, I am happy to recommend the publication of this paper in Molecules. The authors revised the whole manuscript, and emphasize the content to the synthesis of the pyrimidine contained compounds as Aurora kinase and polo like kinase inhibitors. Generally the whole paper contain useful information for the audience who is working on the field of Aurora kinase and polo like kinase, thus I am happy to recommend the publication of this paper.
Author Response
Point 1- English language and style are fine/Minor spell check required.
Response 1- The MS has been revised and spelling errors have been addressed to the best of our ability.
The reviewer has recommended publication of the MS in its current form, we appreciate this.